# Comparative Analysis of Age, Sex, and Viral Load in Outpatients during the Four Waves of SARS-CoV-2 in A Mexican Medium-Sized City

**DOI:** 10.3390/ijerph19095719

**Published:** 2022-05-08

**Authors:** Carlos Eduardo Covantes-Rosales, Victor Wagner Barajas-Carrillo, Daniel Alberto Girón-Pérez, Gladys Alejandra Toledo-Ibarra, Karina Janice Guadalupe Díaz-Reséndiz, Migdalia Sarahy Navidad-Murrieta, Guadalupe Herminia Ventura-Ramón, Mirtha Elena Pulido-Muñoz, Ulises Mercado-Salgado, Ansonny Jhovanny Ojeda-Durán, Aimée Argüero-Fonseca, Manuel Iván Girón-Pérez

**Affiliations:** 1Laboratorio Nacional de Investigación Para la Inocuidad Alimentaria (LANIIA) Unidad Nayarit, Universidad Autónoma de Nayarit, Tepic 63000, Nayarit, Mexico; carlos.covantes@uan.edu.mx (C.E.C.-R.); wagner.laniia@uan.edu.mx (V.W.B.-C.); daniel.giron@uan.edu.mx (D.A.G.-P.); gladys_alejandrat@hotmail.com (G.A.T.-I.); karinajgdr@gmail.com (K.J.G.D.-R.); snavidad.laniia@gmail.com (M.S.N.-M.); herminia.ventura@uan.edu.mx (G.H.V.-R.); mir.pulido96@gmail.com (M.E.P.-M.); ulises.mercado.salgado@gmail.com (U.M.-S.); ansonny.ojeda@uan.edu.mx (A.J.O.-D.); 2Laboratorio de Psicofisiología y Conducta, Unidad Académica de Ciencias Sociales, Universidad Autónoma de Nayarit, Tepic 63000, Nayarit, Mexico; aimee.arguero@uan.edu.mx

**Keywords:** SARS-CoV-2, viral load, age groups, sex, qRT-PCR

## Abstract

Governments have implemented measures to minimize SARS-CoV-2 spread. However, these measures were relaxed, and the appearance of new variants has prompted periods of high contagion known as waves. In Mexico, four waves distributed between July and August 2020, January and February 2021, August and September 2021, and January and February 2022 have appeared. Current health policies discourage mass sampling, preferring to focus on the corrective treatment of severe cases. Outpatients are only advised to undergo brief voluntary confinement and symptomatic treatment, with no follow-up. Therefore, the present study aimed to analyze sex, age, and viral load in outpatients during the four waves in a medium-sized city in Mexico. For each wave, the date of peak contagion was identified, and data were collected within ±15 days. In this regard, data from 916 patients (434 men and 482 women) were analyzed. The age range of positive patients (37–45 years) presented a higher frequency during the first and third waves, while 28–36 years was the most frequent age range during the second and fourth waves, while the viral load values were significantly higher, for both sexes, during the fourth wave. Obtained data of COVID-19 prevalence in population segments can be used for decision-making in the design of effective public health policies.

## 1. Introduction

Coronavirus disease-19 (COVID-19), produced by the severe acute respiratory syndrome coronavirus 2 (SARS-CoV-2), has become a global pandemic and a threat to public health [1]. The symptomatology of SARS-CoV-2 is highly variable; however, common symptoms are observed in other infectious diseases. Patients exhibit fever, dry cough, loss of smell and/or taste, gastrointestinal complaints, headache, muscle pain, and chest pain, and in severe cases, respiratory distress and inability to speak or move, or even death [2,3,4,5].

Globally, countries have implemented restrictive and sanitary measures to control the rate of infection, such as the use of masks, lockdown, social distancing, the control of people’s capacity, and the restriction of business hours [6]; however, months of the continuing pandemic has caused these measures to be relaxed, leading to COVID-19 outbreaks or waves [7]. Since the COVID-19 outbreak in Hubei, China, many countries have presented a multiple wave pattern [8,9,10]. In the particular case of Mexico, epidemiological trends in this country have revealed a four-wave pattern of laboratory-confirmed COVID-19 cases distributed between July and August 2020, January and February 2021, August and September 2021, and January and February 2022. On 4 April 2022, the official data showed that the number of confirmed COVID-19 cases worldwide reached 492,053,168, of which 5,666,215 were from Mexico [11].

Socioeconomic, demographic, and environmental parameters influence the risk of infection. In this context, there are differences between the age groups of infected people among the different epidemiological peaks of SARS-CoV-2 [12]. Within the population groups that are at constant risk of contagion are economically active people (young people). Although some individuals in this sector can work remotely (home office), most workers, given the nature of employment, are required to continue to work in person at the workplace [13]. Moreover, differences between sexes have been reported, due to the fact that there is a male-biased mortality in patients diagnosed as carriers of SARS-CoV-2 [14,15]. Therefore, looking at the effects of the virus on patients of different ages and sexes, it is possible to take appropriate measures for the identification and protection of vulnerable groups [16]. Without underestimating the biological essence of the virus, the viral load data is highly pertinent, so there must be surveillance of this parameter. The viral load is essential to understanding the viral dynamics; patients with severe cases had significantly lower cycle threshold values (Ct) than those with mild cases at the time of admission [17].

Sanitary authorities in Mexico only prioritized data collection related to positive cases, recovered cases, hospitalizations, and deaths. On the other hand, outpatients having little or no follow-up were generally diagnosed by rapid tests (antibodies and antigens) and receive symptomatic treatment. In Mexico, efforts have been made to immunize the population; however, there is no homogeneity in vaccination, vaccination schedules are incomplete, and the available vaccines are mixed. Additionally, there is little transparency in the official data. Therefore, the similarities and differences between sex, age distribution, and viral load (cycle threshold) in outpatients from the four waves of COVID-19 remain largely unknown. Therefore, the present study aimed to compare the characteristics of qRT-PCR-confirmed positive outpatients in Tepic, Nayarit, a medium-sized city in Mexico.

## 2. Materials and Methods

### 2.1. Study Design and Participants

Outpatients presenting for SARS-CoV-2 qRT-PCR testing in our laboratory (LANIIA-UAN) (laboratory approved by the Mexican health authorities (number: DGE-DDYR-DSAT05140-2020)) were prospectively evaluated. All COVID-19-positive outpatients diagnosed by our laboratory during the four waves of COVID-19 in the Mexican state of Nayarit were selected; it should be noted that the laboratory has been performing these tests with the same trained personnel since the beginning of the pandemic. Official data revealed that the peaks of the four waves of SARS-CoV-2 occurred on the following dates: 1 August 2020, 21 January 2021, 19 August 2021, and 19 January 2022; after which, all data (age, sex, and Ct levels) were recorded ±15 days from the date of the peak levels of the four waves. Before conducting this study, all patients were duly informed, gave their consent for use of their data for scientific purposes, and signed an informed consent form. This study was approved by the authorities of the Local Bioethics Commission under registration number CEBN/03/20.

### 2.2. Swabbing

For sampling, patients underwent nasopharyngeal/oropharyngeal swabbing. Both swabs were placed against the mucosa and gently rotated for a few seconds, then removed while rotating and placed in 2.5 mL of sterile VTM. Sterile VTM (pH 7.10) was prepared with Hank’s balanced salt solution (HBSS) (Thermo Fisher Scientific, Cleveland, OH, USA, Cat no. 14190) supplemented with gentamicin sulfate (4 mg/mL) (Thermo Fisher Scientific, Cat no. 15750078), penicillin/streptomycin (50.000 U/50,000 mg/mL) (Thermo Fisher Scientific) (Cat no. 15140148), amphotericin B (0.4 mg/mL) (Thermo Fisher Scientific) (Cat no. 15290018), and bovine serum albumin (5%) (Thermo Fisher Scientific) (Cat no. 15561020).

### 2.3. qRT-PCR Procedure

The method for the molecular diagnosis of SARS-CoV-2 was determined and validated by the Mexican health authorities (Secretaría de Salud de México and InDRE). First, for inactivation and extraction of the total RNA, the commercial QIAmp Viral RNA Mini Kit (Qiagen, Cat No./ID: 1020953 USA, Germantown) was used. The samples used were vortexed, swabs squeezed, and 140 µL of VTM was taken. Then, qRT-PCR was performed according to the Berlin protocol with modifications (Corman et al. 2020) using the StarQ One-Step qRT-PCR kit (Qiagen, Cat No./ID: 210210, USA, Germantown). For SARS-CoV-2 detection for the viral gene, E_Sarbeco_Forward: ACAGGGTACGTTAAGTTAATAGCGT, E_Sarbeco_Reverse: ATATTGCAGCAGCAGTACGCACACACA, and TaqMan probe E_Sarbeco_P1: FAMCACTAGCCATCCTTAGCCTTCCTTCG-BBQ, while the RNAseP gene was used as a control (RNAseP Forward: AGATTTGGACCTGCGAGCG, RNAseP Reverse, GAGCGCTGTCTCCACAAGT, and TaqMan RNAseP P1 probe FAMTTCTCTGAC-CTGAAGGCTCTCTCTCTCTCTG CGCG-BHQ1) [18,19]. The qRT-PCR was performed with 5 µL (~70 ng/µL) of extracted RNA in a total reaction of 25 µL. All samples were analyzed with the ABI Prism 7500 sequence detector system (Applied Biosystems) with the following protocol: 50 °C for 15 min; 95 °C for 2 min; and then 45 cycles of 95 °C for 15 s, 82 °C, and 60 °C for 30 s. In all cases, amplification of the human gene (RNAseP) was used as the internal control, and samples were considered positive if the number of cycles required for the fluorescent signal to cross the threshold cycle, known as the cycle threshold (Ct) value, was equal to or less than 38.

### 2.4. Statistical Analysis 

Age and viral load data were analyzed by the Kruskal–Wallis one-way nonparametric ANOVA test, with Dunn’s post hoc test, using SigmaPlot^®^ 11.0 statistical software (Systat Software Inc., San Jose, CA, USA) with a significance level of *p* < 0.05.

## 3. Results

A total of 916 outpatients (482 women and 434 men), distributed over the four waves of COVID-19 (July 2020–February 2022), were analyzed. All participants were diagnosed as SARS-CoV-2-positive.

First, the age and sex distributions of patients in the four epidemiological waves of COVID-19 in Tepic, Nayarit, Mexico were observed. The data indicated that only the female group during the third wave presented better ages (*p* < 0.05); on the other hand, male patients showed a homogeneous mean age during the four waves of COVID-19 (Table 1). Then, to observe which age groups had the highest prevalence of positive outpatients during the COVID-19 waves, the diagnosed positive cases were visualized in age histograms for both sexes during the four waves of COVID-19. From the histogram, we can see that the young adult age groups (both sexes) were the majority of the outpatients (Figure 1A–C). The analysis of the cycle threshold (Ct) data, which highlights the viral load (inversely proportional values), was also evaluated. A significant reduction of the cycle threshold was observed during the fourth wave of COVID-19 (Figure 2A); this phenomenon was observed for both sexes (Figure 2B,C).

## 4. Discussion

Since the outbreak of severe acute respiratory syndrome coronavirus 2 (SARS-CoV-2), the virus responsible for the disease called COVID-19 in Wuhan, China, the world has faced multiple epidemiological waves [20]. Particularly, in Mexico, the official data has indicated a four-wave pattern, In the case of Nayarit, the data trend displayed four epidemiological waves whose dates corresponded to 1 August 2020, 21 January, 19 August 2021, and 17 January 2022 [11]; then, data 15 days before and after the peak pandemic dates during the four waves of SARS-CoV-2 from all positive outpatients were collected (15 July–14 August 2020, 6 January–5 February 2021, 4 August–3 September 2021, and 3 January–2 February 2022).

Mexican health authorities only report positive cases, recovered cases, hospitalizations, and deaths, focusing primarily on hospitalized patients, but do not efficiently process these data [21]. Combined with poor crisis management, disregard for scientific information, and negligent policies by the national highest authorities, adverse effects are exerted on the design of policy responses [22]. In addition, health services are insufficient, and accessibility to medical units is not equal for different sectors of the population [23,24]. The current health policies discourage mass sampling [25], preferring to focus on the corrective treatment of severe cases. On the other hand, outpatients receive little or no follow-up, are generally diagnosed by rapid tests (antibodies and antigens), receive symptomatic treatment, and only are advised to undergo brief voluntary confinement and symptomatic treatment. Therefore, this ends up affecting, minimizing, and making invisible the effects suffered by outpatients.

The background demographics of SARS-CoV-2-positive outpatients such as age, sex, and viral load were compared. The data analysis found significant differences in age only in females during the four waves (Table 1). Looking at the age distribution of the positive cases, outpatients categorized in the age ranges of 28–36 and 37–45 years (economically active young adults) were those with the highest incidences in the four waves of COVID-19 in Tepic City (Figure 1), a phenomenon reported in several European countries, Saudi Arabia, and the United States, related to behavioral aspects such as a negligent attitude towards health measures, perception of low risk, and social and working lives of young people [26,27,28,29,30]. Additionally, another factor to consider is the kind of job carried out by the majority of this sector of the population, since, in many cases, it is practically impossible to only work from a home office.

The viral load value has been associated with the incubation period [31] and lung damage [32], as well as with severity and fatal outcomes [33,34,35] and, recently, in milder cases, with reduced gray matter thickness, tissue damage in regions functionally connected to the primary olfactory cortex, and an overall reduction in brain size [36]. In the present study, a significantly higher viral load (lower Ct values) was found in the fourth wave of COVID-19 in Nayarit, a medium-sized city in Mexico (Figure 2). Possibly, this higher viral load is due to the fact that, in Nayarit, the delta variant and its subtypes were predominant during the third wave, while omicron and its subvariants were predominant during the fourth wave, according to genomic surveillance [37]. The omicron variant is the most transmissible form of SARS-CoV-2 known so far and evolved predominantly worldwide intermittently [38]. This increased transmissibility is linked to the viral load [39], as could be observed during the fourth wave (Figure 2). High levels of viral RNA caused by infection with the omicron variant have been reported [40]. Sequencing is a useful tool, as it allows the detection of predominant variants in each region or in a certain time range; variant detection is very relevant, as the mutations present in each variant may represent a potential escape strategy from the immune response. Likewise, the presence of variants also has an impact on the differential symptomatology that exists between them, so there are variants of higher risk than others [41]. In line with the above, the peaks of cases in the third and fourth waves may be attributable to the delta (B.1.617.2) and omicron (B.1.1.529) variants; both variants have higher transmissibility, showing an increased ability to infect cells and evade the immune response compared to the native variant [42,43,44,45,46,47].

The most reliable method for the detection of severe acute respiratory syndrome coronavirus-2 (SARS-CoV-2) is RT-PCR [5]. This technique is expensive and inaccessible to most people; the advances in vaccination have compensated in some way for these deficiencies, since the vaccination of high-risk groups has reduced the mortality rate and viral shedding [47,48,49,50]. Vaccination is the key to halting the spread of the SARS-CoV-2 virus; however, the production, distribution, and application of the vaccines to all segments of the population is a challenging task [6]. Mexico is trying to immunize its population; however, there is no homogeneity in the vaccinations, the vaccination schedules are incomplete, and the available vaccines are mixed; added to this, it is difficult to obtain official data. This is reflected in the local scenario in the city of Tepic, the vaccines CanSino Biologics Inc. (Beijing, China); Pfizer -BioNTech (Germany), AstraZeneca (Oxford, UK), Sinovac Biotech (Beijing, China), Moderna (Massachusetts, United States) and Johnson and Johnson (Janssen Pharmaceuticals, Beerse, Belgium were applied by using incomplete, complete, and mixed schedules. In addition, the data availability is outdated for vaccine coverage. Despite the fact that all currently available vaccines are effective against all detected variants of concern, people do not trust the quality and efficacy of the vaccines, so they choose not to get vaccinated and take home remedies in the case of infection.

The pandemic caused by COVID-19 reached its maximum peak during the fourth wave, in spite of the fact that the vaccination process was already in the application phase of the second dose in individuals under 20 years of age and the booster dose (third dose) in some population groups, such as individuals over 40 years of age and teachers. Due to different factors, such as the relaxation of sanitary protocols and the appearance of COVID-19 variants, as well as the refusal to vaccinate in the state of Nayarit, may cause the temporary lengthening of the pandemic and be a risk factor for the development of new mutations and variants of the SARS-CoV-2 virus, which may modify the efficacy of the vaccinations, compromising the ability to contain the SARS-CoV-2 pandemic.

## 5. Conclusions

The age range of ambulatory patients (28–36 and 37–45 years) in both sexes presented a higher frequency during the COVID-19 waves; this is the economically active sector of the population. On the other hand, the viral load was similar between the first three waves of COVID-19; however, during the fourth wave, the viral load was significantly higher. Health policies should consider pertinent actions aimed at young adults to avoid chains of contagion derived from the high activity of this group, even though the mortality rate is relatively low. In addition, given the high mobility of young adults, they indirectly interact on a daily basis with individuals belonging to vulnerable groups, representing a risk of contagion. These data are important, as they can be used for decision-making and in the design of effective public health policies. Such data inform the prevalence of COVID-19 in segments of the population, which can focus on successful preventive strategies and inform young people and restrict activities where large numbers of this group congregate.

## Figures and Tables

**Figure 1 ijerph-19-05719-f001:**
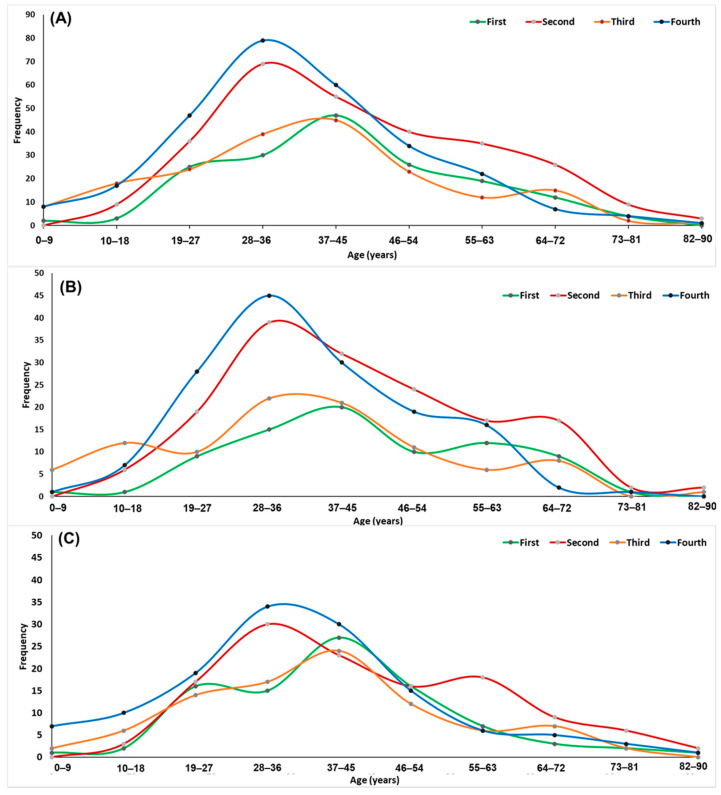
Age–frequency distribution of outpatients diagnosed as SARS-CoV-2 carriers sorted by COVID-19 waves in the city of Tepic, Mexico (**A**): age distribution of female patients (**B**) and male patients (**C**).

**Figure 2 ijerph-19-05719-f002:**
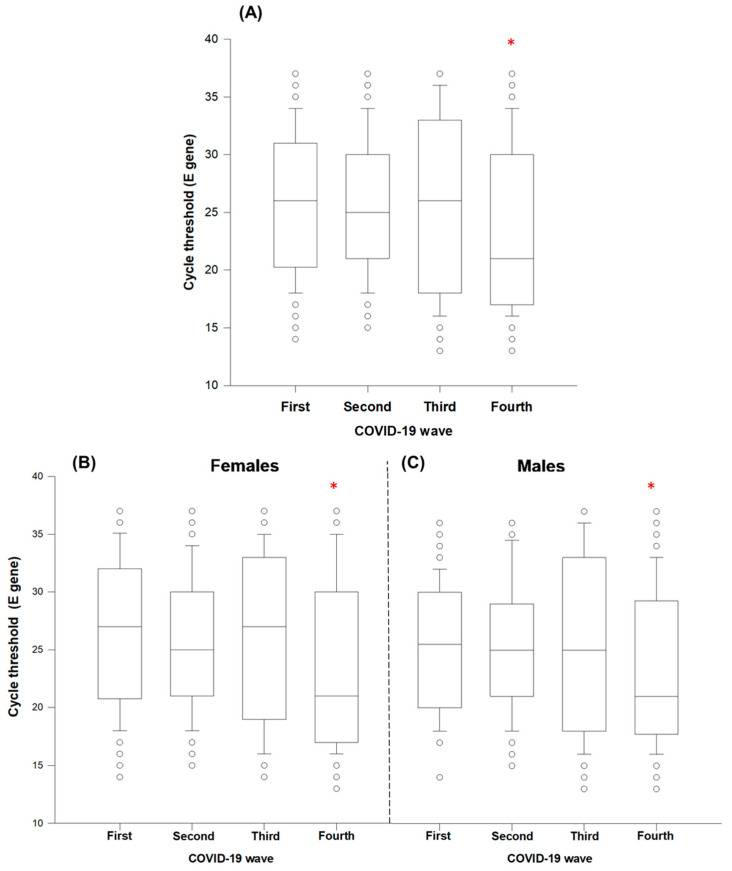
Viral load (cycle threshold) of outpatients among the four waves of COVID-19 reported for Nayarit, Mexico (**A**): viral load of females (**B**) and males (**C**). * Nonparametric Kruskal–Wallis and Dunn’s multiple comparison tests were performed (*p* < 0.05).

**Table 1 ijerph-19-05719-t001:** Mean age by sex of outpatients during the four peaks of the SARS-CoV-2 pandemic in the city of Tepic.

	COVID-19 Wave				
		First	Second	Third	Fourth
Females	N	78	158	97	149
	Age				
	(Mean ± SD)	44 ± 15 a	43 ± 16 a	36 ± 17 b	40 ± 14 ab
	Range (year)	(7–75)	(10–87)	(2–87)	(6–82)
Males	N	90	124	90	130
	Age				
	(Mean ± SD)	41 ± 15 a	44 ± 16 a	39 ± 16 a	39 ± 18 a
	Range (year)	(4–86)	(15–83)	(2–79)	(5–93)

Different letters indicate significant differences between groups. Nonparametric Kruskal–Wallis and Dunn’s multiple comparison tests were performed (*p* < 0.05).

## Data Availability

All data supporting the findings of the study are available with its corresponding author, MIGP, upon reasonable request.

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
