# Peer review of "Comparative Analysis of Age, Sex, and Viral Load in Outpatients during the Four Waves of SARS-CoV-2 in A Mexican Medium-Sized City"

_ijerph, 2022, doi:10.3390/ijerph19095719_

Round 1

Reviewer 1 Report

I believe the idea is well presented as well as the methodology. The interpretation and aim of the study must be more developed. I have mentioned my remarks in the document attached

Reviewer 2 Report

Interesting observation, but it requires additional epidemiological analysis and methodological support:

- It is very descriptive, and it would benefit of a larger consideration of data and confounders. Sex age and/vs viral load are not sufficient to explain an hypothesis and suggest an explanation. It can be a short report, but it would need to be confirmed by more evidences and possibilities for explanations.

- e.g., serological data: what is the immuno-protection of the population in the different waves? Can this explain the differences? What is the vaccination coverage? Also data on differences in lockdown measures and quarantine distribution during/before the different waves may help, as well as data on clinical severity of the diseases measured for example reporting the occupancy in ICU.

- e.g., clinical and pharmacological data: what about the health state changed in the population between the different waves? What kind of lethality was observed and what age? How distributed and who survived in the different ages, potentially modifying the susceptibility in the next wave? What about the social issues claimed in the discussion, the distribution of the social and economical groups for the different ages how was modified in the different waves? Any antiviral drug was used? and in what percentage in the different waves and population ages? Any data of susceptibility groups like smokers asthmatic immunoexpressed etc or on the use of possible/putative protective agents such as lactoferrin of Vit D levels or Vit  or nutritional differences..?

- Regarding the methods, were samples stored for the same time and processed by an identical laboratory and following the same protocol? Swab sampling was performed by the same operator??? (or data compared by same operator and same lab/protocol?). Why is reported only the value for one gene (E) when it is commonly used at least 2 genes and sometimes 4? If other primer set for other viral genes were used, the trend result is comparable with the E pattern? If the human internal control (Pol) was used, why this data were not reported and it was not used to answer at least some of the previous questions..!?! 

- Several genetic differences involved the different waves, including different variants of interest and their distribution in the population. Any data is available on the SARS-CoV-2 V.O.I. diffusing during the considered period, and their relative role in the different pikes?

In conclusion, the proposed observation is valuable and interesting; but it deserves a deeper speculation and needs to be supported by additional information to constrain confounders and allow reader to acquire indications for possible explanations or speculations based on reported evidences.

Round 2

Reviewer 1 Report

I am happy with the answers of the authors. 

I believe the paper is ok right now for the publication.

Reviewer 2 Report

Most of the observations were fulfilled.